

# Hall anomaly and moving vortex charge in layered superconductors

Assa Auerbach[1*] and Daniel P. Arovas[2]

**1** Physics Department, Technion, Haifa, Israel
**2** Department of Physics, University of California at San Diego,
La Jolla, California 92093, USA

⋆ assa@physics.technion.ac.il

## Abstract

Magnetotransport theory of layered superconductors in the flux flow steady state is revisited. Longstanding controversies concerning observed Hall sign reversals are resolved. The conductivity separates into a Bardeen-Stephen vortex core contribution, and a Hall conductivity due to moving vortex charge. This charge, which is responsible for Hall anomaly, diverges logarithmically at weak magnetic field. Its values can be extracted from magetoresistivity data by extrapolation of vortex core Hall angle from the normal phase. Hall anomalies in $YBa_2Cu_3O_7$, $Bi_2Sr_2CaCu_2O_{8-x}$, and $Nd_{1.85}Ce_{0.15}CuO_{4-y}$ data are consistent with theoretical estimates based on doping dependence of London penetration depths. In the appendices, we derive the Streda formula for the hydrodynamical Hall conductivity, and refute previously assumed relevance of Galilean symmetry to Hall anomalies.

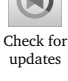

## 1  Introduction

The Hall effect in the flux flow (FF) regime of superconducting films has long been an intriguing and controversial subject. The pioneering theory of Bardeen and Stephen (BS) [1] predicted that the Hall sign is inherited from the normal phase which persists inside the vortex cores. Soon thereafter, in a challenge to BS theory, Hall sign reversals have been measured in diverse superconductors [2–6]. This effect, named "Hall anomaly", is illustrated in Fig. 1. Proposed explanations included the effects of disorder [2], thermal excitations [7], interlayer vorticity [8], and vortex charge [6, 9, 10]. These effects were incorporated into the FF transport theory by vortex dynamics equations [9, 11, 12], and time dependent Ginzburg-Landau theory [13–15].

However, the microscopic origin of *non-dissipative* vortex forces and imaginary relaxation rates, as well as the definition of the relevant vortex charge in a screened environment, have been subjects of ongoing debates, lasting for more than 50 years.

In this paper, we take on the Hall anomaly problem with a new approach to flux flow transport theory. This approach allows us to show that an additional Hall current is carried by moving vortex charge (MVC), which does not vanish in a layered superconductor geometry. The MVC can be estimated from independently measured thermodynamic coefficients, and

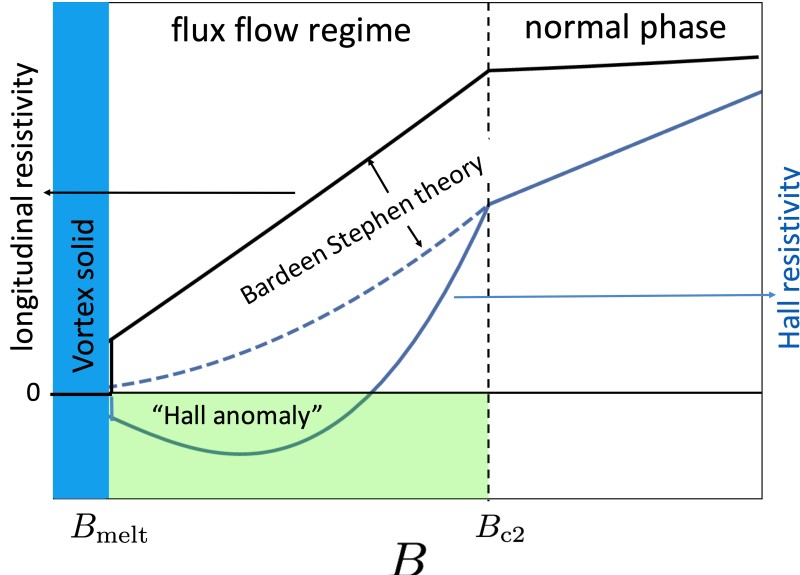

Figure 1: Bardeen Stephen theory and Hall anomaly in the flux flow regime. For this illustration, vortex cores are assumed to obey Drude theory of metals. The deviation of Hall resistivity from Bardeen Stephen theory, and its sign reversal, are ascribed to moving vortex charge.

compared to the value extracted from magnetoresistivities data.

The strategy of this paper is as follows.

(i) We first revisit FF transport theory. While a vortex dynamical equation was used to explain dissipative FF transport [16, 17], its form and some of its coefficients have not been microscopically derived. The source of the difficulty might be in computing the resistivity by imposing a bias current on superconductors without Galilean symmetry (see expanded discussion in Section 9).

Here, in contrast to previous approaches [1–17], we completely avoid bias currents, vortex forces and their phenomenological equations of motion. The mean vortex velocity in the steady state is constrained by the external electric field. The current is calculated as a linear response to the applied electric field, and Galilean symmetry is not assumed.

(ii) We show that the conductivity (per layer) separates into two additive terms,

$$\sigma_{\alpha\beta} = \left(\frac{B_{c2}(T)}{B}\right)\sigma_{\alpha\beta}^{\text{core}} + \epsilon_{\alpha\beta}\frac{2|e|Q_{\text{v}}}{h} \qquad . \tag{1}$$

$e < 0$ is the electron charge, and $\epsilon_{\alpha\beta}$ with $\alpha, \beta \in \{x, y\}$ is the antisymmetric tensor. The first term recovers BS formula [1] which describes the transport currents generated inside the moving vortex cores. $B$ and $B_{c2}$ are the magnetic field and upper critical field respectively. The second term is due to the MVC, whose value is $Q_{\text{v}}$.

(iii) We use the Streda formula [18, 19] to relate $Q_{\text{v}}$ to the extra charge induced into the superconducting layer by the addition of one vortex. In isotropic superconductors, $Q_{\text{v}} = 0$ due to in-plane screening by co-moving charges. In layered superconductors, screening in dopant (weakly superconducting) layers results in,

$$Q_{\text{v}} = Q_0 \log(\alpha B_{c2}/B) \quad . \tag{2}$$

$Q_0$ is proportional to the derivative of superfluid stiffness with respect to electron density, and to the interlayer dielectric constant, which can be experimentally determined without microscopic knowledge of the normal state correlations. $\alpha$ represents vortex core properties and is of order unity. Generically, $Q_0$ has opposite sign to first term in Eq. (1), which can produce the Hall sign reversal depicted in Fig. 1.

(iv) $Q_{\text{v}}$ can be extracted from experimental magnetresistivity data, and compared to Eq. (2), using a Hall angle extrapolation for estimating the BS term. Values extracted for hole doped [6, 20], and electron doped [20] cuprates are consistent with Eq. (2), where $Q_0$ can be fit to independent estimates of the doping derivative of London penetration depth and interlayer dielectric constant.

We briefly discuss effects of inhomogeneous pinning at low magnetic field, and superconducting fluctuations. The paper ends with a summary of our results and their comparison to previous theories.

## 2 Flux Flow Steady State

We consider a homogeneous thin film of a type-II superconductor, below the zero field transition temperature $T < T_{\text{c}}^{(0)}$, A magnetic field $B = B\hat{z}$ induces a two dimensional (2D) vortex

density [1] $n_{\mathrm{v}} = B/\Phi_0 = (2\pi l_B^2)^{-1}$, where $\Phi_0 = hc/(2|e|)$ is the Josephson flux quantum. The FF regime is defined by (see Fig. 1),

$$\max\left\{ B_{\mathrm{melt}}(T), \frac{\Phi_0}{2\pi\lambda^2(T)} \right\} \; < \; B(T) \; < \; B_{\mathrm{c2}}(T), \tag{3}$$

where $B_{\mathrm{c2}} \equiv \Phi_0/2\pi\xi^2$, $\xi$ is the vortex core radius, and $B_{\mathrm{melt}}$ is the vortex lattice melting field [17, 21, 22]. For thin enough films, London's penetration depth can easily exceed the inter-vortex separation $\lambda \gg l_B$. Hence, the magnetic field is approximately uniform and a more appropriate term for "flux flow" would be "vorticity flow".

In cuprates, and other highly anisotropic layered superconductors, $B_{\mathrm{melt}}(T) \ll B_{\mathrm{c2}}(T)$ at low temperatures [23]. In the FF regime defined in Eq. (3), the critical current is zero, or immeasurably small, such that we will later be able to apply linear response theory to compute the longitudinal and transverse conductivities.

The global phase field of the superconductor, outside the vortex cores, can be separated into the vorticity phase and transport phase:

$$\begin{aligned} \phi(\boldsymbol{x}) &= \phi_{\mathrm{v}}(\boldsymbol{x}) + \phi_{\mathrm{tr}}(\boldsymbol{x}) \\ \phi_{\mathrm{v}} &= \mathrm{sgn}(e) \sum_{i=1}^{N_{\mathrm{v}}} \arg(\boldsymbol{x} - \boldsymbol{X}_i), \end{aligned} \tag{4}$$

where $\boldsymbol{X}_i$ are the vortices' positions, and $\oint d\boldsymbol{\ell} \cdot \nabla\phi_{\mathrm{tr}} = 0$ on any orbit.

Henceforth we fix the gauge choice to be $A_0(\boldsymbol{x}, t) = 0$ throughout the paper. For a configuration of static vortices $\dot{\phi}(\boldsymbol{x}) = 0$. We introduce an external DC electric field into the vector potential as $\boldsymbol{A} = \boldsymbol{A}_B - c\boldsymbol{E}t$, where $\nabla \times \boldsymbol{A}_B = B\hat{z}$. If $\phi(\boldsymbol{x})$ remains time-independent, the 2D current density will increase linearly in time, *viz.*

$$\boldsymbol{j}(t) = \frac{2e}{\hbar}\rho_s\left(\boldsymbol{\nabla}\phi - \frac{2e}{\hbar c}\boldsymbol{A}(t)\right) \propto \boldsymbol{E}\,t \quad, \tag{5}$$

where $\rho_s$ is the 2D superfluid stiffness, and the mean free energy density will increase as $f(t) \sim \rho_s t^2$. The runaway energy will be cut off by destruction of the superfluid stiffness, or by mobilization of the vortices, which will result in $\boldsymbol{\nabla}\dot{\phi} \neq 0$.

Let us first consider a single moving vortex with velocity $\boldsymbol{V}$ as depicted in Fig. 2. Outside the vortex core of radius $\xi$, a dipolar electromotive force field (EMF) $\mathcal{E} = -\frac{\hbar}{2e}\boldsymbol{\nabla}\dot{\phi}_{\mathrm{v}}$, is associated with the vortex motion. The EMF inside the metallic vortex core $\mathcal{E}^{\mathrm{core}}$ is determined by the voltage drop between the core boundary points at $\partial_\xi = \{\boldsymbol{x} \,|\, |\boldsymbol{x} - \boldsymbol{X}|^2 = \xi^2\}$,

$$\int_{\boldsymbol{x}_1}^{\boldsymbol{x}_2} d\boldsymbol{\ell} \cdot \mathcal{E}^{\mathrm{core}} = \frac{\hbar}{2e}\left.\dot{\phi}_{\mathrm{v}}\right|_{\boldsymbol{x}_2} - \frac{\hbar}{2e}\left.\dot{\phi}_{\mathrm{v}}\right|_{\boldsymbol{x}_1}, \tag{6}$$

where $\boldsymbol{x}_i \in \partial_\xi$. From the definition of $\phi_{\mathrm{v}}(\boldsymbol{x} - \boldsymbol{V}t)$, the core EMF is linearly related to the vortex velocity by,

$$\mathcal{E}^{\mathrm{core}} = \frac{\hbar}{2|e|\xi^2}\hat{z} \times \boldsymbol{V}. \tag{7}$$

Since $\nabla^2\phi_{\mathrm{v}} = 0$ everywhere, $\mathcal{E}$ is divergence free, in analogy to an in-plane 2D "magnetic field". The EMF produced by each moving vortex can be parameterized by a 2D "magnetic moment",

$$\boldsymbol{\mu}_i = \frac{\xi^2}{2}\mathcal{E}_i^{\mathrm{core}} \quad. \tag{8}$$

---

[1] In this paper, a vortex (antivortex) is defined by the anticlockwise (clockise) circulation of the electrical current, rather than the winding of the phase (see Figure 2).

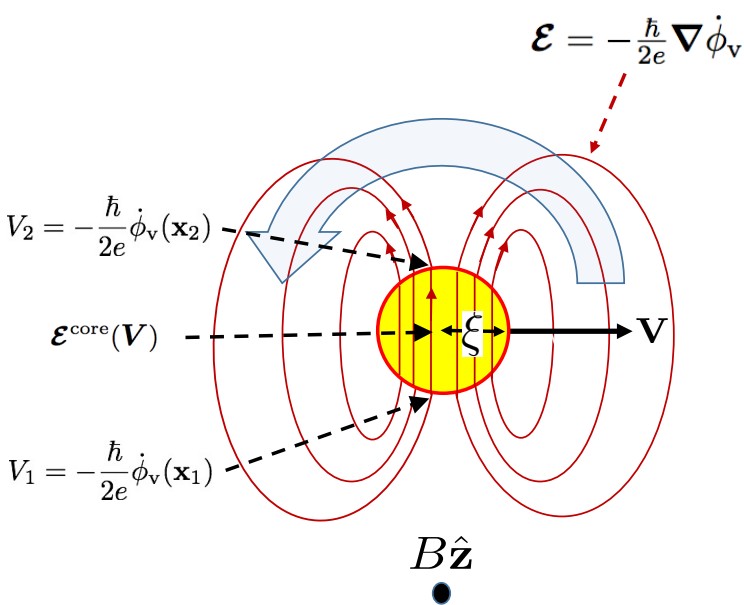

Figure 2: Electromotive force field (EMF) $\mathcal{E}$ (red lines with arrows), created by a single moving vortex of velocity $V$. The circulating supercurrent is depicted by a wide blue arrow. A core EMF, $\mathcal{E}^{\text{core}}$, is imposed by the voltage drop $(V_1 - V_2)$ across the metallic region (yellow disk) of radius $\xi$.

In analogy with magnetostatics [24], a finite density of 2D "moments" produces a 2D "magnetization field" $m = n_{\text{v}} \bar{\mu}$. Henceforth $\bar{o} \equiv \frac{1}{\Delta A} \int_{\Delta A} d^2x \, o$, denotes coarse graining of $o$ over an area $\Delta A$ which includes many vortices. The relation between the EMF and these "magnetic moments" is given by the "magnetic field" to "magnetization" ratio, $\bar{\mathcal{E}} = 4\pi m$. By Eqs. (7-8) this relation implies,

$$\bar{\mathcal{E}} = \frac{h n_{\text{v}}}{2|e|} \, \hat{z} \times V \equiv \frac{h}{2|e|} \hat{z} \times \mathcal{J}_{\text{v}} = -V \times B/c \quad .\tag{9}$$

$V$ denotes the average vortex velocity, and $\mathcal{J}_{\text{v}}$ is the vorticity current. The last equality in Eq. (9) uses $n_{\text{v}} = B/\Phi_0$, and is known as Josephson's relation [25]. In vortex dynamics approaches [1,16], Josephson's relation is used to express the EMF as a function of the computed vortex velocity.

Here we turn this relation on its head. By demanding a steady state $\frac{d}{dt}\langle j \rangle = 0$, the externally imposed electric field must be cancelled, on average, by $-\frac{\hbar}{2e}\langle \nabla \dot{\phi} \rangle$, and hence $E = \bar{\mathcal{E}}$. This constrains the average vortex velocity to be,

$$V(E) = \frac{c}{B^2} E \times B.\tag{10}$$

It should be emphasized that in our approach, the vortex velocity is independent of any Hamiltonian parameters. The transport problem is formulated in terms of a linear response of the transport current to the externally applied electric field $E$.

The current has two separate components: the metallic vortex cores which serve as ideal current pumps, and the MVC transported by the moving vortices (see Fig. 3).

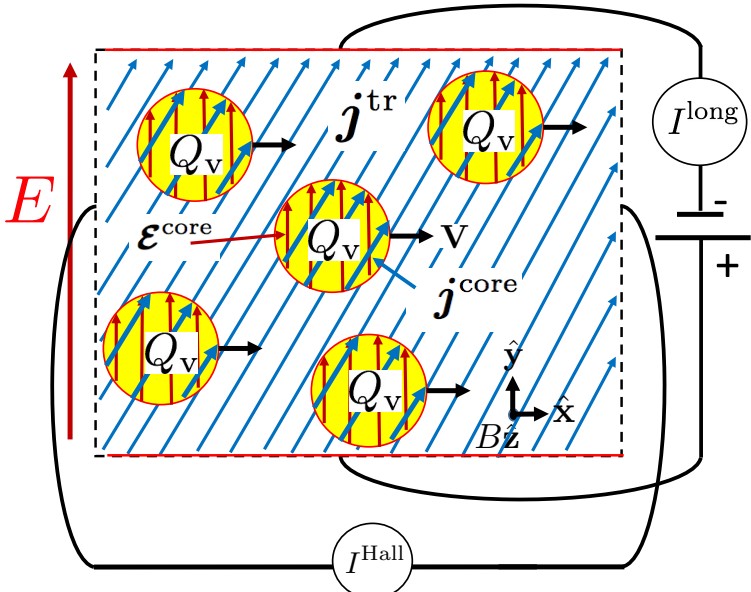

Figure 3: Flux flow steady state transport. The electric field dictates the mean vortex velocity according to Josephson relation $V = \frac{c}{B^2} E \times B$. Each moving vortex core is an ideal current pump of both longitudinal and Hall components $j^{\text{core}}$. The core currents combine to a transport current density $j^{\text{tr}} = j^{\text{core}}$ marked by long blue arrows. Moving vortex charges of average magnitude $Q_{\text{v}}$ produce an additional Hall current, which is responsible for the Hall anomaly.

## 3 Core and transport currents

Within the BS model, the vortex cores are metallic disks of radius $\xi$, with superconducting stiffness $\rho_{\text{s}} = 0$ inside the core and constant outside. The electrons in the disk are subjected to a core EMF which interpolates between its boundaries which move at velocity $V$ of Eq. (10), as depicted in Fig. 2. The scattering impurities are, naturally, at rest in the lab frame. Lorentz transformation of the EMF from the moving frame to the lab frame (to linear order in $V/c$) combined with the external field $E$, yields $\mathcal{E}^{\text{core}} + E + V \times B/c = \mathcal{E}^{\text{core}}$, where we have used Eq. (10). The core current in the lab frame is determined by the core conductivity tensor,

$$j_\alpha^{\text{core}} = \sum_\beta \sigma_{\alpha\beta}^{\text{core}} \mathcal{E}_\beta^{\text{core}}. \tag{11}$$

For a homogeneous superconducting film, we assume that all vortices move at the same average speed and produce the same core current density. The moving vortex cores act as *ideal current pumps* of the DC transport current. Due to the finite viscosity of the core electrons, the current density is continuous at the core boundaries,

$$j^{\text{core}} = j^{\text{tr}}(x) \equiv \frac{2e}{\hbar} \rho_{\text{s}} \nabla \phi_{\text{tr}}(x) \qquad (x \in \partial_\xi) \quad , \tag{12}$$

where $\phi_{\text{tr}}$ was defined in Eq. (4). For weak enough electric field, additional vortex-antivortex pairs in the superconducting medium are not produced by the vortex motion and the transport current remains laminar, *i.e.* $\nabla \times \nabla \phi_{\text{tr}} = 0$. By charge conservation, $\nabla^2 \phi_{\text{tr}} = 0$ everywhere outside the vortex cores. Thus, as a harmonic function, $\phi_{\text{tr}}$ is determined (up to a constant) by its gradients on the vortex core boundaries. The unique solution for the transport current, is a globally uniform current density $j^{\text{tr}} = j^{\text{core}}$, as depicted in Fig. 3.

Local fluctuations in core currents and charge density are eliminated in the coarse grained DC current density $\bar{\boldsymbol{j}}^{\mathrm{tr}}$. Substituting Eq. (7) in Eq. (11), the BS conductivity is obtained:

$$\sigma_{\alpha\beta}^{\mathrm{BS}} = \left(\frac{B_{c2}(T)}{B}\right)\sigma_{\alpha\beta}^{\mathrm{core}} \quad, \tag{13}$$

which constitutes the first term in Eq. (1).

Note that the BS Hall angle $\tan(\theta_{\mathrm{H}}^{\mathrm{BS}}) = \sigma_{xy}^{\mathrm{BS}}/\sigma_{xx}^{\mathrm{BS}}$, equals to the Hall angle of the metallic core. In Drude metals, $\tan(\theta_{\mathrm{H}}) = \omega_{\mathrm{c}}\tau_{\mathrm{tr}}$, where $\omega_{\mathrm{c}}$ is the cyclotron frequency and $\tau_{\mathrm{tr}}(T)$ is the transport relaxation time which varies on the temperature scales of the normal phase. The alternative Hall angle result of Nozieres and Vinen [11] is not validated by the derivation above.

In the "dirty limit" $\xi \gg l_{\mathrm{tr}}$ [2], where $l_{\mathrm{tr}}$ is the core mean free path, $\sigma^{\mathrm{core}}(T,B)$ can be approximated by extrapolating the normal phase conductivity $\sigma^{\mathrm{normal}}(T,B)$ from $T > T_{\mathrm{c}}(B)$ to $T < T_{\mathrm{c}}(B)$. For cuprate superconductors and other cases of relatively short $\xi$, an extrapolation procedure which exploits the continuity of the BS Hall angle will be proposed in Section 7.

Fig. 1 illustrates the expected BS resistivities for Drude-theory core conductivity. The additional effect of MVC is discussed in the following sections.

## 4 Hall conductivity of charged vortices

In this section we discuss the contribution of MVC to the Hall conductivity. In the static $\boldsymbol{E}=0$ case, in the absence of micrscopic particle-hole symmetry, one expects local vorticity currents to induce charge density modulations $\delta\rho^{\mathrm{vc}}(\boldsymbol{x}-\boldsymbol{X}_i)$, which would be centred around the vortex positions $\{\boldsymbol{X}_i\}$. Vortex charge has been previously proposed in the context of Hall anomalies [9,10], see discussion in Section 9. Vortex induced charge density modulations have also been observed experimentally [26]. If indeed each vortex drags (on average) an MVC of value $Q_{\mathrm{v}}$, there will be an additional Hall current given by,

$$\boldsymbol{j}^{\mathrm{mvc}} = Q_{\mathrm{v}}\mathcal{J}_{\mathrm{v}} = \frac{2|e|Q_{\mathrm{v}}}{h}\boldsymbol{E} \times \hat{\boldsymbol{z}}, \tag{14}$$

which will produce the second term of Eq. (1). (See Fig. 3).

The sign and magnitude of $Q_{\mathrm{v}}$ are a-priori open to many options: The total conduction electron density per vortex or some fraction of it [15]? The superconducting condensate density per vortex? Coulomb screening could neutralize the total vortex charge, and must be carefully considered [27]. These dilemmas have long been debated.

Here, MVC is well defined using Kubo linear response theory. In the case of $\lambda \gg l_B$, the vortex system behaves as an incompressible fluid due to the long range logarithmic interactions between vortices. The coarse-grained charge density response to any local variation of the magnetic field $\delta B_{\mathrm{v}}$ is given by

$$\delta\bar{\rho}^{\mathrm{vc}}(\boldsymbol{x},t) = \int d^2x' \, dt' \, R(\boldsymbol{x}-\boldsymbol{x}',t-t') \, \delta B_{\mathrm{v}}(\boldsymbol{x}',t'), \tag{15}$$

where, by the aforementioned incompressibility, $R$ is a local function of space and time.

The dynamical Hall conductivity of the charged vortex fluid is proportional to the Fourier transform of $R$ [19],

$$\sigma_{xy}^{\mathrm{mvc}}(\boldsymbol{q},\omega) = c\tilde{R}(\boldsymbol{q},\omega), \tag{16}$$

---

[2]For simplicity we ignore proximity and Andreev reflection corrections at the vortex core boundaries, which would be small in the dirty limit of $\xi \gg l_{\mathrm{tr}}$.

which is shown in Appendix A. The second term in Eq. (1) is the DC limit $\sigma_{xy}^{\mathrm{mvc}} = \lim_{\omega \to 0} \sigma_{xy}^{\mathrm{mvc}}(0, \omega)$.

Since $R$ is a local function in space and time, $\tilde{R}(q, \omega)$ is a smooth function of $q, \omega$, whose order of limits $(q, \omega) \to 0$ commute. Therefore, we can reverse the order of limits, and obtain the thermodynamic relation,

$$\sigma_{xy}^{\mathrm{mvc}} = c \lim_{q \to 0} \tilde{R}(q, 0) = c \left( \frac{\partial \bar{\rho}^{\mathrm{vc}}}{\partial B} \right)_\mu \equiv \frac{2|e|}{h} Q_{\mathrm{v}}, \tag{17}$$

where the first equality is known as the Streda formula [18] (see Appendinx A). Eq. (17) defines $Q_{\mathrm{v}}$ as

$$Q_{\mathrm{v}} \equiv \frac{d}{dN_{\mathrm{v}}} \int d^2 x \, \bar{\rho}^{\mathrm{vc}}(x), \tag{18}$$

that is to say, $Q_{\mathrm{v}}$ is the change in total charge after inserting one additional flux quantum into the system. For the calculation in the following section, we note that the background charge density which is not created by the vorticity, does not contribute to the MVC Hall current.

Streda formula is known for its application to gapped quantum Hall (QH) phases. There, the order of limits $(q, \omega) \to 0$ also commute due to the locking of local charge and magnetic flux variations. Intriguingly, the local flux-charge attachment property is shared between the QH and the vortex liquid phases [28]. The difference between the vortex fluid and the QH liquid is that $Q_{\mathrm{v}}$ in the QH phases is quantized at particular rational multiples of $e$.

# 5 Screened Ginzburg-Landau theory of the MVC

Three dimensional screening ensures that the total charge accumulated around a static vortex vanishes, except near the surface [27]. Here we need to know whether for a moving vortex, the screening charges move with the vortex and cancel any contribution of $\delta \rho^{\mathrm{vc}}$ to the Hall current.

This question is answered by applying the theory of vortex charge screening of Khomskii and Freimuth [9] to the layered superconductor. Since Thomas-Fermi screening length is much shorter than $\xi$ and $l_B$, the local electrochemical equilibrium equation,

$$e \, \delta \varphi(r) + \delta \mu(r) = 0, \tag{19}$$

relates between between the screening electrostatic potential $\varphi$, and the local chemical potential deviation $\delta \mu$ induced by the vorticity. Here, $r = (x, z)$ is a three dimensional (3D) coordinate. The 3D charge density deviation is determined by Poisson's equation,

$$\nabla^2 \varphi(r) = -\frac{4\pi}{\epsilon_0} \delta \rho^{\mathrm{3D}}(r), \tag{20}$$

where $\epsilon_0$ is the local dielectric constant. Our goal is now to determine the profile of $\delta \mu(r)$.

In the absence of pinning, our vortex fluid is described as a slowly flowing hexagonal vortex lattice (VL). The quantities calculated will be accurate to leading order in the vortex velocity. We begin with the 2D Ginzburg-Landau (GL) free energy density of the superconducting condensate,

$$f = a |\Psi|^2 + \tfrac{1}{2} b |\Psi|^4 + K \left| (\boldsymbol{\nabla} - \tfrac{2ie}{\hbar c} \boldsymbol{A}) \Psi \right|^2, \tag{21}$$

with $\boldsymbol{A} = \frac{1}{2} B \hat{\boldsymbol{z}} \times x$. In the superconducting phase, $a < 0$ and the coherence length is $\xi = (-K/a)^{1/2}$. Throughout we assume $B \ll B_{\mathrm{c2}}$, in which case we may write $\Psi(x) \approx (-a/b)^{1/2} \exp(i\phi)$, where $\phi$ is given by Eq. (4) after setting $\phi_{\mathrm{tr}} = 0$. This form

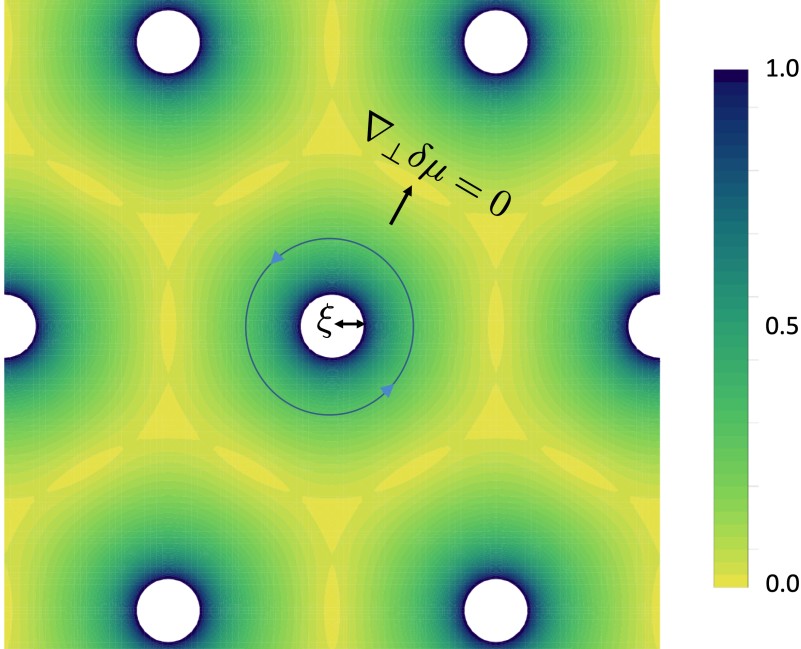

Figure 4: Ginzburg Landau free energy density of a vortex lattice, $f^{\mathrm{vl}}(x,y)$ of Eq. (22), in arbitrary units. $\xi$ is the vortex core radius, and blue arrows represent a circulating current. The local chemical potential deviation $\delta\mu(x) = df^{\mathrm{vl}}/dn_{\mathrm{e}}$, is approximately proportional to $f^{\mathrm{vl}}(x,y)$. By symmetry, the normal gradient $\nabla_{\perp}\delta\mu$ vanishes on the unit cell boundaries. If the total charge per unit cell is non zero, it must be screened in the third dimension, as depicted in Fig. 5

for $\Psi(x)$ presumes that the vortex core size is zero, which we shall correct below. The GL free energy density is thus written as

$$f^{\mathrm{vl}}(x) = \frac{\rho_{\mathrm{s}}}{2}\left\{-\frac{1}{2\xi^2} + (\nabla\Phi)^2\right\} + \rho_{\mathrm{s}}\frac{\varepsilon_{\mathrm{core}}}{\xi^2}\sum_{i}\Theta(\xi - |x - X_i|), \tag{22}$$

where $\rho_s = -2Ka/b$ is the superfluid stiffness, $\varepsilon_{\mathrm{core}}$ is the dimensionless vortex core energy, and

$$\Phi(x) = \frac{\pi}{2}\frac{B}{\Phi_0}|x|^2 - \sum_{i}\ln|x - X_i| \quad . \tag{23}$$

Setting $\langle\nabla^2\Phi\rangle = 0$ (charge neutrality of a 2D Coulomb gas) forces the vortex density to be rigidly determined by $n_{\mathrm{v}} = B/\Phi_0$. The profile of $f(x)$, outside the vortex cores, is depicted in Fig. 4.

The local chemical potential deviation in Eq. (19) can be derived from Eq. (22),

$$\delta\mu(x, 0) = \frac{df^{\mathrm{vl}}}{dn_{\mathrm{e}}}(x). \tag{24}$$

In isotropic three dimensional superconductors, $\partial_z\delta\mu(x, z) = 0$. The in-plane screening case is depicted in Fig. 5(a). The total 2D charge deviation in the unit cell area,

$$\int_{\mathrm{uc}} d^2x \; \nabla_{\perp}^2\delta\mu(x, z) = 0, \tag{25}$$

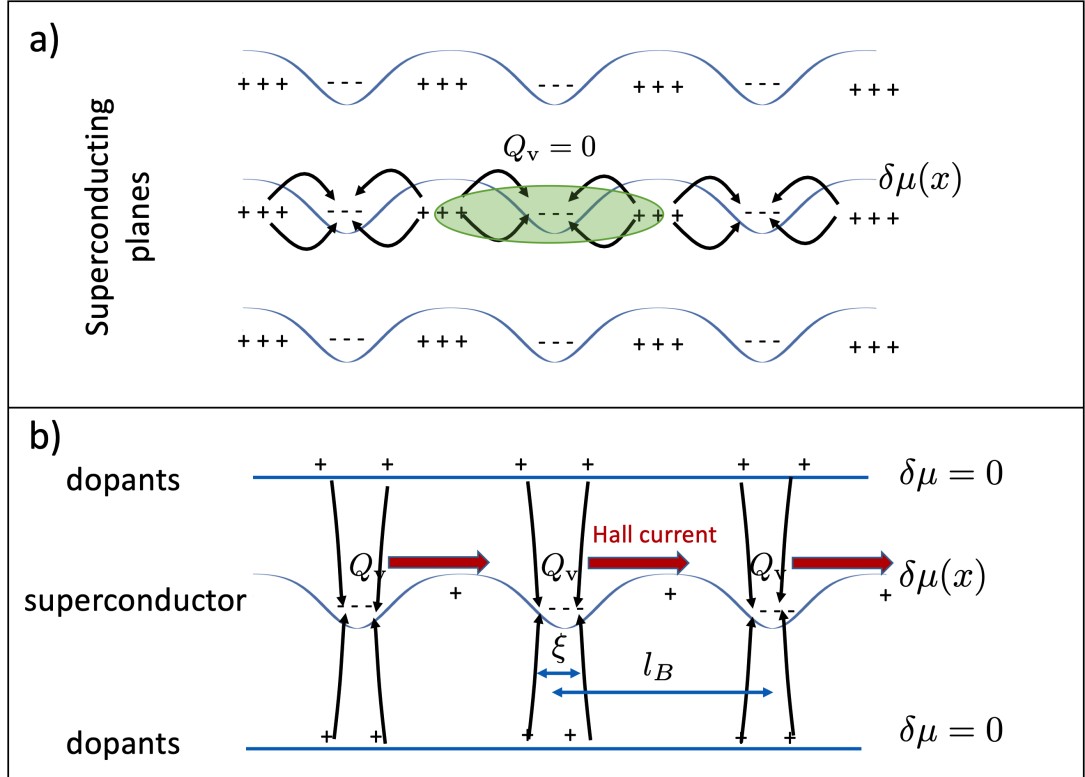

Figure 5: Three dimensional screening of vortex charge density. (a) Stacked superconducting planes with in-plane screening. (b) Staggered dopant layers and superconducting planes where an unscreened Hall current flows in the superconducting planes.

which is the consequence of $\vec{\nabla}_{\perp}\delta\mu^{\mathrm{vl}}(r)$ vanishing by symmetry on the 2D unit cell boundaries, see Fig. 4.

In layered superconductors, neighboring dopant layers are positioned at $z = \pm a_c/2$, with $\delta\mu(x, \pm a_c/2) \propto \rho_s^{\mathrm{dop}} \ll \rho_s$. The interlayer modulation can be approximately modelled by,

$$\delta\mu^{\mathrm{vl}}(x,z) = \frac{1}{2}\frac{df^{\mathrm{vl}}}{dn_e}(x)\Big(1 + \cos(2\pi z/a_c)\Big) + \mathcal{O}(\rho_s^{\mathrm{dop}}/\rho_s) \; . \tag{26}$$

This modulation results in interlayer screening. The charge deviation given by Eq. (20) is

$$\delta\rho^{\mathrm{3D}}(x,z) = \frac{\epsilon_0}{4\pi e}\left\{\nabla_{\perp}^2 - \left(\frac{2\pi}{a_c}\right)^2\right\}\delta\mu^{\mathrm{vl}}(x,z) \quad . \tag{27}$$

The MVC, by (17), is given by differentiating the total areal charge of a superconducting plane, with respect to vortex number:

$$Q_{\mathrm{v}} = \frac{d}{dN_{\mathrm{v}}}\int d^2x \int_{-a_c/4}^{a_c/4} dz\, \rho^{\mathrm{3D}}(x,z) = -\frac{\epsilon_0}{2ea_c}\frac{d}{dN_{\mathrm{v}}}\frac{d}{dn_e}\int d^2x f^{\mathrm{vl}}(x) \quad . \tag{28}$$

We can discard the uniform condensation energy $-\rho_s/4\xi^2$ in Eq. (22) which does not depend on $N_{\mathrm{v}}$. As shown in Eq. (25), the integral over the in-plane Laplacian $\nabla_{\perp}^2$ vanishes over each

unit cell. The relevant vortex charge density is obtained by the second contribution of Eq. (27). This charge density is screened on the neigboring dopant layers, see Fig. 5(b).

Averaging Eq. (22) over a unit cell (UC) of the vortex lattice can be performed analytically [29],

$$Q_{\mathrm{v}} = -\frac{\epsilon_0 \pi}{2 e a_c} \frac{d}{d n_{\mathrm{e}}} \left\{ \rho_{\mathrm{s}} (\varepsilon_{\mathrm{M}} + \varepsilon_{\mathrm{core}}) \right\} \quad , \tag{29}$$

where $\varepsilon_{\mathrm{M}}$ is the dimensionless Madelung energy,

$$\varepsilon_{\mathrm{M}} = \frac{1}{2} \ln \left( \frac{B_{\mathrm{c2}}}{\sqrt{3}\pi B} \right) - \ln |\eta(\tau)|^2 \quad , \tag{30}$$

where $(1, \tau = \exp(i\pi/3))$ are the complexified triangular lattice vectors, $q = \exp(2i\pi\tau)$ and

$$\eta(\tau) = |q|^{1/24} \prod_{n=1}^{\infty} (1 - q^n) \quad , \tag{31}$$

is the Dedekind eta function. The dimensionless vortex core energy $\varepsilon_{\mathrm{core}}$ is of order unity in BCS theory [30]. The $\log(1/B)$ dependence in Eq. (30) results from integration of the vorticity current squared, $|x - X_i|^{-2}$, over a unit cell area $2\pi l_B^2$.

Combining Eqs. (29-31), we arrive at a compact formula for the MVC in terms of the GL parameters,

$$Q_{\mathrm{v}}(T, B) = Q_0 \log(\alpha B_{c2}/B) \quad , \tag{32}$$

where the temperature dependent parameters are,

$$Q_0(T)/e = -\frac{\pi \epsilon_0}{4 e^2 a_c} \cdot \frac{d \rho_{\mathrm{s}}}{d n_{\mathrm{e}}} \tag{33}$$

$$\log \alpha(T) = -1.47 + \frac{2}{\pi} \varepsilon^{\mathrm{core}} - \rho_{\mathrm{s}} \left( \frac{d \rho_{\mathrm{s}}}{d n_{\mathrm{e}}} \right)^{-1} \left( \frac{\partial B_{c2}}{\partial n_{\mathrm{e}}} B_{c2}^{-1} - \frac{2}{\pi} \frac{\partial \varepsilon^{\mathrm{core}}}{\partial n_{\mathrm{e}}} \right) \,.$$

$\alpha(T)$ is of order unity, and depends on $\xi$, and $\varepsilon^{\mathrm{core}}(n_{\mathrm{e}})$. These quantities require a microscopic theory of the core properties, but do not effect the value of $Q_0$ and the logarithmic dependence of $Q_{\mathrm{v}}$ on magnetic field in Eq. (32).

# 6 Extraction of $Q_{\mathrm{v}}$ from experiment

One would like to extract MVC values from experimental Hall and longitudinal magnetoresistivities. The problem is that in unconventional superconductors, we often do not fully understand the behavior of the metallic core conductivities, which are required for the subtraction. Fortunately, the Hall conductivity in Eq. (1) exhibits a separation between the core and the MVC contributions which can be exploited.

The BS Hall angle inside the metallic cores reflects the extrapolated behavior of the normal phase. For temperatures not too far from $T_{\mathrm{c}}$, it is reasonable to linearly extrapolate of the normal state temperature dependences of $\rho_{xx}, \rho_{yx}$ to below $T_{\mathrm{c}}$, as depicted in Fig. 6. This yields an extrapolated values of $\tan \theta_{\mathrm{H}} \to \tan \bar{\theta}_{\mathrm{H}}$.

Thus, the BS Hall conductivity can be deduced by multiplying the measured $\sigma_{xx}^{\mathrm{exp}}$ (which is not affected by the MVC), by the extrapolated Hall angle,

$$\sigma_{xy}^{\mathrm{BS}} = \sigma_{xx}^{\mathrm{exp}}(B, T) \tan \bar{\theta}_{\mathrm{H}}(B, T). \tag{34}$$

Using Eq. 1, we can extract the experimental values of the MVC by subtracting $\sigma_{xy}^{\mathrm{BS}}$ from the experimental Hall conductivity $\sigma_{xy}^{\mathrm{exp}}$,

$$Q_{\mathrm{v}}^{\mathrm{exp}}(B, T) \equiv \frac{h}{2|e|} \left( \sigma_{xy}^{\mathrm{exp}} - \sigma_{xy}^{\mathrm{BS}} \right). \tag{35}$$

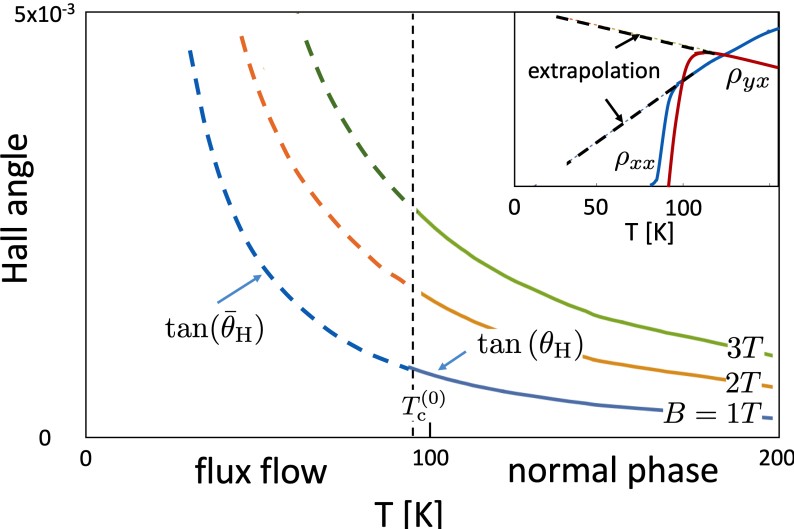

Figure 6: Extrapolation (dashed lines) of the Hall angle from the normal state of $Bi_2Sr_2CaCu_2O_{8-x}$ into the FF regime using the ratio of linearly extrapolated resistivities (Inset) of the normal phase data of Ref. [6].

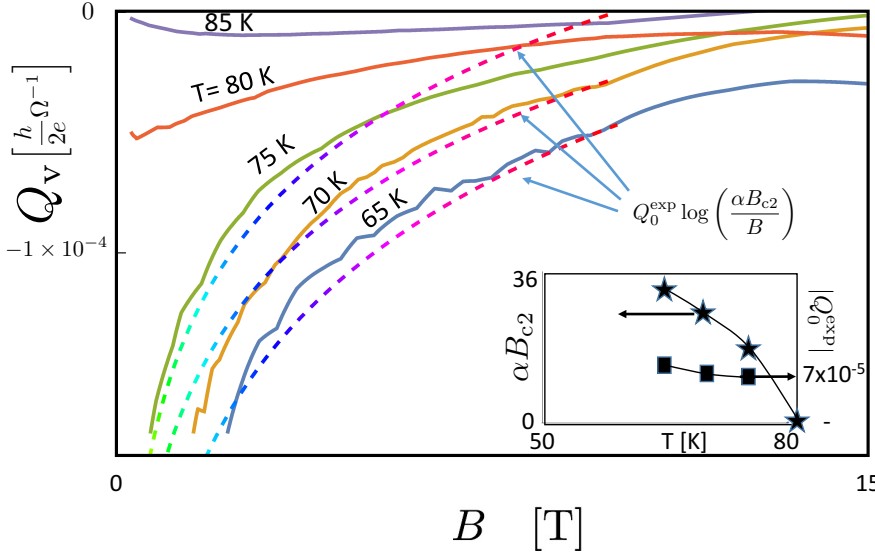

Figure 7: Moving vortex charge $Q_v$ extracted from Zhao *et al.* [6]. The resistivity data of two unit cell thickness film of $Bi_2Sr_2CaCu_2O_{8-x}$ are inputs into the procedure described in Section 7. Dashed lines are fits to theory, Eq. 32. Inset: Fitted parameters $\alpha B_{c2}$ and $Q_0^{exp}(T)$, (which is negative). The weak temperature dependence of $Q_0^{exp}$ is consistent with a constant parameter $\gamma$ which is roughy consistent with phenomenological Uemura's relations [31, 32] of Eq. (37).

# 7 MVC of cuprate superconductors

A crude theoretical estimation of $Q_0$ of Eq. (33), can be obtained from measured doping and temperature dependent London penetration depth $\lambda$, in bulk three dimensional cuprate su-

Table 1: Vortex charge coefficients of cuprates. Using Eq. 35, $Q_0^{\text{exp}}$ is extracted from Hall and magnetoresistivity data at fixed temperature $T[\text{K}]$. $Q_0^{\text{th}}$, of Eq. 39, is fit to $Q_0^{\text{exp}}$ using the listed values of $\epsilon_0^{\text{fit}}$. Data from refs. [6, 20, 34, 35]. See text for details.

| Compounds | $T[\text{K}]$ | $Q_0^{\text{exp}}/e$ | $\lambda/a$ | $Q_0^{\text{th}}/e$ | $\epsilon_0$ |
|---|---|---|---|---|---|
| $Bi_2Sr_2CaCu_2O_{8-x}$ | 75 | −0.45 | 684 | $-0.05\epsilon_0$ | 9 |
| $YBa_2Cu_3O_7$ | 91 | −1.68 | 410 | $-0.13\epsilon_0$ | 12.4 |
| $Nd_{1.85}Ce_{0.15}CuO_{4-y}$ | 11 | +0.97 | 533 | $+0.086\epsilon_0$ | 11.3 |

perconductors. The two dimensional (per layer) superfluid stiffness $\rho_s$, is related to $\lambda$ by

$$\rho_s = \frac{\Phi_0^2}{4\pi^3}\frac{a_c}{\lambda^2} = 6.3 \times 10^{-4}\,\lambda^{-2} \times a_c\,[\text{eV/cm}] \quad . \tag{36}$$

$a_c$ is the $c$-axis layer separation, which will later drop out of $Q_0$.

In the underdoped regime $x \leq 0.15$, the doping and temperature dependent $\rho_s$ of cuprates [31–33] is roughly captured by an empirical formula

$$\rho_s(x, T) = \rho_s(x_{\text{opt}}, 0)\left(\frac{|x|}{x_{\text{opt}}}\right) - \gamma k_B T\,, \tag{37}$$

for $|x| < x_{\text{opt}}$, where $x = 1 - n_e a^2$ is the doping concentration per copper, $x_{\text{opt}} \approx 0.16$ is optimal doping, and $a \simeq 3.8$Å. is the copper-copper distance in the superconducting planes. Note that $x$ is positive (negative) for hole (electron) doped materials, and that $\gamma$ is weakly doping and temperature dependent, which is consistent with the empirical Uemura scaling [31,32], given by $\rho_s^{(0)} \approx \gamma T_c$.

Thus,

$$\frac{d\rho_s(T)}{dn_e} \simeq -\text{sgn}(x)\,a^2\,\frac{\rho_s(x_{\text{opt}}, 0)}{|x_{\text{opt}}|}\,, \tag{38}$$

which by Eqs. (33) and (36) yields,

$$Q_0^{\text{th}} = -3650\,e\,\text{sgn}(x)\left(\frac{\epsilon_0}{x_{\text{opt}}}\right)\left(\frac{a}{\lambda}\right)^2\,, \tag{39}$$

whose values for hole-doped $Bi_2Sr_2CaCu_2O_{8-x}$ and $YBa_2Cu_3O_7$, and electron-doped $Nd_{1.85}Ce_{0.15}CuO_{4-y}$, are listed in Table 1.

The experimental values of $Q_0^{\text{exp}}$ in Table 1, are extracted from the data of Refs. [6, 20]. The agreement is quite reassuring: we can fit $Q_0^{\text{exp}} = Q^{\text{th}}$ using $\epsilon_0 = 9 - 11.3$. This short wavelength parameter is difficult to obtain experimentally. Dielectric constants are expected to be similar in the different cuprates, due to similar local environments. These values are not extremely different from $\epsilon_0 = 4.5$ which was used to fit ellipsometry data of $Bi_2Sr_2CaCu_2O_{8-x}$ in Ref. [36].

## 8 Inhomogeneous flow and fluctuations

This paper has implicitly assumed weak effects of disorder and pinning in the FF regime. Eq. (1) applies to homegenous vortex motion, 'deep' in the FF regime as defined by Eq. (3).

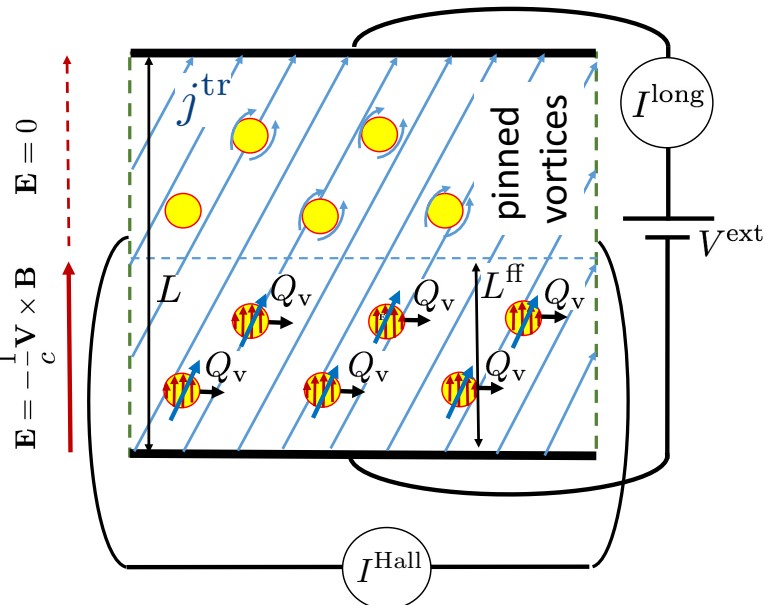

Figure 8: Inhomogeneous flux flow. The pinned vortices do not contribute to the voltage drop, and the transport currents avoid their metallic cores. The external voltage $V^{\text{ext}}$ drops only on the flux flow strip of width $L^{\text{ff}}$, which effectively enhances the Bardeen Stephen conductivity by a factor of $L/L^{\text{ff}}$, but not the moving vortex charge contribution, see Eq. (40).

Short range (relative to $\xi$) disorder determines the normal state conductivities of $\sigma^{\text{core}}$. Long range disorder may broaden the melting transition at weak fields, due to formation of "vorticity rivers" of total cross section $L^{\text{ff}}$, between pinned regions of cross section $L - L^{\text{ff}}$. In Fig. 8 we depict two domains, which can be generalized to describe realistic systems with multiple rivers and pinned domains.

The FF conductivity, Eq. (1) is readily modified to take into account such inhomogeneous flow. The mean vortex velocity in the rivers, is enhanced by a geometric factor $V \to V\frac{L}{L^{\text{ff}}}$. Thus, the core currents of the moving vortices are also enhanced by that factor. Due to the laminar inter-vortex current flow, the identity $j^{\text{tr}} = j^{\text{core}}$ for the average current density still holds. The transport current through the superconductor bypasses the resistive cores of the pinned vortices, see Fig. 8. The BS conductivity is therefore multiplied by a factor of $L/L^{\text{ff}}$.

The MVC Hall current, however, remains unchanged, because it is proportional to the total vorticity current, and the effective conductivity due to partial pinning is

$$\sigma_{\alpha\beta}^{\text{FF−partial}} = \left(\frac{B_{\text{c2}}}{B}\right)\left(\frac{L}{L^{\text{ff}}}\right)\sigma_{\alpha\beta}^{\text{core}} + \epsilon_{\alpha\beta}\frac{2|e|Q_{\text{v}}}{h}. \tag{40}$$

The partial pinning parameter $L^{\text{ff}}/L < 1$ depends on the sample inhomogeneities, and is an increasing function of field, temperature, and current. A signature of the partial pinning regime, would be a non linear current-voltage relation.

We note partial pinning enhances both the BS Hall and longitudinal conductivities. Therefore, we can still extract $Q_{\text{v}}^{\text{exp}}$ from $\sigma_{xx}^{\text{exp}}, \sigma_{xy}^{\text{exp}}$, and the extrapolated Hall angle, using Eq. (35). The primary difficulty at low fields is to measure the extremely low values of $\rho_{xx}$ and $\rho_{xy}$ in the linear response regime.

Fig. 1 depicts the resistivities of Bardeen-Stephen mean field theory, in which $B_{\text{c2}}$ is a sharp phase transition. In reality, for quasi-2D films and especially for few monolayers as

in Ref. [6], $T_c(B)$ (or $B_{c2}(T)$) denote a broad crossover between the flux flow and normal metal regimes. Evidently, short range phase correlations which are required to define vortices, persist in a limited range of $T > T_c$, and $B > B_{c2}(T)$. In the thin $Bi_2Sr_2CaCu_2O_{8-x}$ films, $T_c = T_{KT}$ is of Kosterlitz-Thouless type, where, as Halperin and Nelson [37] have shown, mobile vortices exist in some range above $T_{KT}$. Consequently, if these vortices are charged by $Q_v \neq 0$, they can produce a Hall sign reversal even slightly above $T_{KT}$. In thicker films, superconducting order pairing fluctuations [17,38] play an important role above $T_c$. We note that for cuprates, the normal state (a.k.a. the "bad metal") longitudinal and Hall conductivities are poorly understood.

## 9 Discussion

Hall anomalies in superconductors have been an open theoretical problem for more than four decades. Some of the confusion in the field originates in the formulation of vortex forces and viscosities. Vortex dynamics approaches calculate the vortex velocity (and the associated EMF it generates) as a response to the force exerted by a bias current, in addition to other forces. The conductivity tensor depends on the coefficients of these forces. However, conductivities can only be methodically computed by Kubo formulas, where the current is derived as a response to an applied electric field (as in our approach), and not vice versa.

The vortex dynamical equation has seen endless debates since first introduced by BS [1] and Nozieres and Vinen [11] to describe flux flow. Missing in this equation are effects of realistic band structure, impurities, time-retardation and the multi-vortex interactions. While the friction term can be derived from the longitudinal core conductivity by equating the Joule power dissipation [1], non-dissipative forces have been difficult to justify microscopically [39]. Hall voltage producing forces were related to imaginary relaxation time of the time dependent Ginzburg Landau equation [13], and to the topological phase-density action [15]. The sign and magnitude of such a terms were determined by Ref. [15] by appealing to Galilean invariance. However, this argument is not valid to flux flow in superconductors for two fundamental reasons:

1. Superconductors in a periodic potential with impurities are not Galilean invariant. Kelvin's circulation theorem is valid for a perfectly Galilean liquid, where vortices must move at the velocity of the background current, *i.e.* "go with the flow". This rule would indeed produce the Galilean Hall conductivity of $\sigma_{xy} = c\rho/B$. However, de Gennes and Nozieres [40] pointed out that superconducting vortices may only exhibit "go with the flow" effects for $\omega_c \tau = \tan(\theta_H) \gg 1$. As seen in Fig. 6, the experimental values of $\tan(\theta_H) \sim 10^{-3}$ are well outside this regime. A counterexample is provided around half filling $\rho a^2 \simeq 1$ ($a$ is a lattice constant), where the $\sigma_{xy}$ vanishes and changes sign due to particle hole symmetry of the bandstructure.

2. As shown by several arguments in Appendix B, the onset of short range superconducting stiffness prevents the vortices from "going with the flow". The vortex liquid is incompressible due to its long range interactions. As a consequence, the Hall conductivity is given by the non Galilean invariant Streda formula, Eq. (46): $\sigma_{xy} = c(\partial\rho/\partial B) \neq c\rho/B$.

In this paper the vortex velocity is not driven by various forces, but *dictated* by the electric field via Josephson relation (10). The moving vortices' cores act as ideal transport current pumps. The derivation of Eq. (1) vindicates BS result for the magnetic field dependent Hall conductivity (in the absence of MVC), *vis-a-vis* Nozieres and Vinen's [11] Galilean symmetry motivated prediction.

An important feature of the present approach is the separability of the conductivity tensor in Eq. (1). This feature allows us to subtract the core conductivity term, which is poorly understood in many systems. The normal state conductivities are highly sensitive to material specific disorder, superconducting fluctuations (discussed in Section 8) , and inelastic scattering. In cuprates, it is fair to say, little is yet understood about its normal state transport coefficients.

Hall sign reversals are a result of $Q_v \sigma_{xy}^{\mathrm{core}} < 0$. Whether there are one or two sign reversals, depends on the individual temperature dependence of each of the two terms in Eq. (1). $\sigma_{\alpha\beta}^{\mathrm{core}}$ depends on quasiparticle scattering time, while the MVC is determined by the superfluid stiffness which saturates at low temperatures. A second (lower temperature) sign reversal, such as observed in Ref. [6], indicates that quasiparticle scattering is dominated by inelastic (electron-electron and electron phonon) processes.

Previous authors [9, 10] have discussed the role of vortex charge in Hall anomalies, with emphasis on the vortex core charge density. Coulomb screening however can completely neutralize the total vortex charge in an isotropic superconductor, as depicted in Fig. 5(a). Here we have considered the geometry of dopant layers with interlayer screening. This allows the vortices to drag the MVC within the superconducting layers, while the vertically displaced screening charges reside in neighboring low mobility dopant layers. The major contribution for the MVC arises from vorticity kinetic energy density far outside the cores. Integration of this contribution over inter-vortex separation yields the $\log(B_{c2}/B)$ dependence of $|Q_v|$ at low fields, which is consistent with the experimental data in Fig. 7.

Eq. (1) has been applied to analyse cuprate flux flow transport data in Section 7. We find reassuring agreement between estimates of the doping-dependent London penetration depth, interlayer dielectric constant, and the values of $Q_0$ extracted from experiments, as given in Table 1. Independent measurements of vortex charging in these systems by resonance, capacitance and force microscopy, would be very informative.

At very strong magnetic fields, Hall sign reversals in cuprates have been attributed to Fermi surface reconstructions in the normal metallic phase [41,42], which exhibits quantum oscillations. It would be interesting to investigate the crossover between Hall anomalies at relatively weak field magnetic fields, and Fermi-liquid regimes at much higher fields.

## Acknowledgements

We thank Amit Kanigel, Bert Halperin, Michael Reznikov, and Ari Turner, for useful input and discussions. We also thank Referees 1 and 2 for urging us to clarify our stand on aspects of previous theoretical work. We acknowledge support from US-Israel Binational Science Foundation grant 2016168, Israel Science Foundation grant 2021367, Aspen Center for Physics, Grant NSF-PHY-1066293, where parts of this work were done, and the workshop at ICTS/topmatter2019/12.

## A  Dynamical Hall conductivity and Streda formula

We restrict ourselves to a 2D system with area $A$. The dynamical Hall conductivity is defined as

$$\sigma_{xy}(\omega) = \frac{\hbar}{AZ} \sum_{n,m} \left( \frac{e^{-\beta E_n} - e^{-\beta E_m}}{(E_m - E_n)(E_m - E_n - \hbar\omega)} \right) \mathrm{Im}\left( \langle n|j_q^x|m\rangle \langle m|j_{-q}^y|n\rangle \right), \qquad (41)$$

where $Z = \mathrm{Tr}\, e^{-\beta H}$, and $\{E_n, |n\rangle\}$ is the spectrum of $H$.

Charge conservation yields,

$$\dot{\rho}_q = \frac{i}{\hbar}[H, \rho_q] = -i q \cdot j_q. \tag{42}$$

Maxwell's equation $j = c \nabla \times \mathbf{M}$, relates the magnetization to the current density, and in Fourier representation, $j_q = ic\, q \times \mathbf{M}_q$. Without loss of generality, we can choose $q = (q_x, 0, 0)$ and $M = M^z \hat{z}$, and relate the matrix elements,

$$\langle n|j_q^x|m\rangle = -\frac{1}{\hbar q_x}(E_n - E_m)\langle n|\rho_q|m\rangle$$

$$\langle m|j_{-q}^y|n\rangle = icq_x\langle m|M_{-q}^z|n\rangle. \tag{43}$$

Inserting the current matrix elements (43) in (41) yields

$$\sigma_{xy}(q, \omega) = \frac{c}{AZ} \sum_{n,m} \left( \frac{e^{-\beta E_n} - e^{-\beta E_m}}{(E_m - E_n - \hbar\omega)} \right) \mathrm{Re}\left( \langle n|\rho_q|m\rangle\langle m|M_{-q}^z|n\rangle \right)$$

$$\equiv c\frac{\delta\rho}{\delta B}(q, \omega). \tag{44}$$

The DC Hall conductivity is given by the DC transport limit,

$$\sigma_{xy} = \lim_{\omega \to 0} \lim_{q \to 0} \sigma_{xy}(q, \omega). \tag{45}$$

The Streda formula is the static thermodynamic susceptibility,

$$\sigma_{xy}^{\mathrm{Streda}} = \lim_{q \to 0} \lim_{\omega \to 0} \sigma_{xy}(q, \omega) = c\left(\frac{\partial\rho}{\partial B}\right)_{\mu, T}. \tag{46}$$

When the limits $(q, \omega) \to (0, 0)$ of Eq. (44) commute, $\sigma_{xy} = \sigma_{xy}^{\mathrm{Streda}}$. This condition is satisfied in gapped quantum Hall phases, and for the vortex liquid discussed in Section 4. In both of these systems the response of $\delta\rho(x, t)$ to $\delta B(x', t')$ defined by $R(x - x', t - t')$ in Eq. (15) is local. For resistive gapless metals, the two limits are different, and Streda's formula does not describe the Hall conductivity.

## B  Invalidity of Galilean Hall conductivity in flux flow regime

To discuss the hydrodynamic contribution to the Hall current we consider "coreless" vortices by setting $\sigma_{xy}^{\mathrm{core}} \to 0$ in Eq. (1). Here we do not assume underlying particle hole symmetry, since we wish to discuss the approximately Galilean invariant superconductor.

In a normal Galilean invariant liquid with total charge density $\rho$, vortices "go with flow" as dictated by Kelvin's circulation theorem. That is to say the averaged particles' velocity should be equal to that of the vortices, *i.e.* $j_{\mathrm{Hall}} = \rho V$. In the presence of an electric field, Josephson's mean vortex velocity is $V = c\mathbf{E} \times \hat{z}/B$ – see Eq. (10) – and the Galilean invariant (GI) Hall conductivity would be

$$\sigma_{xy}^{\mathrm{GI}} = \frac{c\rho}{B}. \tag{47}$$

In this Appendix, we show that Eq. (47) is not valid in the flux flow regime of superconductors even in an approximate Galilean Hamiltonian. The conceptual point is that Eq. (47) does not capture the effects of short range superconducting stiffness $\rho_s \neq 0$, which differentiates between flux flow and normal metal regimes.

1. The current induced by a moving vortex in the (Galilean invariant) dynamical Gross-Pitaevskii theory was calculated in the superfluid phase by Arovas and Freire [43] using the mapping to dual (2+1)D electrodynamics:

$$\mathbf{j}_v \propto \frac{\hat{z} \times (\mathbf{r} - Vt)}{|\mathbf{r} - Vt|^2} + \mathcal{O}(V^2). \tag{48}$$

This current integrates to zero, and does not produce a global Hall current $j_{\text{Hall}} = \rho V$, which is required for Eq. (47).

2. Consider a charged two dimensional bosonic superfluid of density $n$ and charge $q$, and Hamiltonian

$$H = \sum_i \frac{(p_i - \frac{q}{c}A)^2}{2m^*} \tag{49}$$

on a narrow ring, subjected to a perpendicular magnetic field $A = -\frac{B}{2}r \times \hat{z}$. A vortex lattice of density $n_v = B/\Phi_0$ is produced in the bulk of the ring. Now a radial electric field $E \parallel \hat{r}$ is adiabatically turned on. The vortex lattice will start moving around the annulus at Josephson's velocity $V = cE \times \hat{z}/B$. We can understand the null drag effect of the moving vortex lattice by analogy to that of moving potential term $\int d^2x \, \varphi(x - Vt)\rho(x)$.

At low velocities, the rigid condensate "sticks" to the lab frame in which it was prepared, and is not dragged by the moving potential (in other words: it is hard to pump a superfluid). This statement can be verified by the following *Gedankenexperiment*: we boost the Hamiltonian to the rotating frame of velocity $V$, which is implemented by inserting an effective Aharonov-Bohm (AB) flux $\Phi_{AB} = (m^*c/q)VL$. Superconducting stiffness ensures that the circulating persistent current in the moving frame would be $j = -nqV$ at a finite flux. The persistent current does not decay even in the presence of the (now) static potential $\varphi(x)$. (This is to be contrasted with the metallic phase, where the current decays to zero by scattering at all fluxes $\Phi_{AB} = \text{integer} \times \Phi_0$.) Boosting back to the non-rotating lab frame, by setting $\Phi_{AB} \to 0$, restores the motion of the potential, and rewinds the current back to zero. Thus the moving potential does not drag the condensate.

3. Due to the local stiffness (rigidity) of the superconductor, the vortex liquid is incompressible due to the logarithmic interactions between vortices. This justifies the interchange of $q, \omega$ order of limits of the hydrodynamical Hall conductivity, resulting in the applicability of the Streda formula $\sigma_{xy} = c(\partial \rho/\partial B)$. The Streda formula for the vortex liquid is proven in Appendix A. Thus, the distinction between a Galilean normal metal and a lower temperature flux flow regime can be captured by the inequality of $\rho/B \neq (\partial \rho/\partial B)$.

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
