# Peer review of "Hall anomaly and moving vortex charge in layered superconductors"

_SciPost Physics, doi:SciPost Phys. 8, 061 (2020)_

## Round 2 · Referee Report · Anonymous · 2020-2-16

Strengths

The manuscript is written in а clear enthralling language and endowed with nice figures hence will constitute a fascinating reading.

Report

The manuscript presents a fascinating phenomenological consideration of the vortex-related Hall effects in superconductors. Although not new, the topic remains alive and attracts a lot of interest since measurements of the Hall effect is an irreplaceable tool for inferring material properties. Thus, an attempt claiming to resolve longstanding controversies of the one of the central phenomenon of the subject, the Hall sign reversal, most certainly worth attention. The clear language and nice pictures will make absorbing reading, so the manuscript may be considered for publication after a couple of minor technical issues will be resolved.

1. The Hall sign reversal phenomenon includes also a double sign change, i.e. the return to the positive sign (if one takes the sign in a normal state as positive) upon further cooling. In particular, this double sign change was observed in Ref. [7] of the manuscript. Can the proposed phenomenological consideration reveal the reason for that and explain this double change?

2. According to the beautiful sketch of Figure 1 of the manuscript, the sign change is expected to occur below Bc2. However, the are many works (including, again, Ref. [7] of the manuscript, which is indeed one of the most recent sign reversal measurements) reporting that the sign change starts already above Bc2. How should the sketch of Figure 1 reconcile with these observations?

3. Reference [7] of the manuscript not only reports measurements of the Hall sign reversal but also presents a quantitative description of the experimental data in the framework of vortex dynamics (Reference [15] of the manuscript). Based on the fair agreement between the theory and experiment, Reference [7] claims that the theory developed in [15] completely explains the phenomenon of the Hall sign change. One could then conclude that Ref. [15] of the manuscript together with the subsequent extension by Geshkenbein, Ioffe, and Larkin, Superconductivity in a system with preformed pairs, PRB 55, 3173 (1997), have already resolved all the controversies existing before. It would be useful for a reader to learn which deficiencies of these (Ref [15] and PRB of 1997) works were resolved in the present manuscript.

4. In calculating the screening effect, the authors apply the theory of the vortex screening by Khomskii and Freimuth (Ref. [10] of the manuscript). However, this theory gives the extra charge of the opposite sign than that obtained by other authors. How does this discrepancy may influence the conclusions of the manuscript?

After these issues are resolved, the paper may be considered for publication.

  • validity: ok
  • significance: good
  • originality: good
  • clarity: good
  • formatting: excellent
  • grammar: excellent

Author:  Assa Auerbach  on 2020-03-06  [id 752]

(in reply to Report 1 on 2020-02-16)
Category:
answer to question

Response to Referee Report 1

We thank the referee for a thorough report, and for urging us to clarify several important points concerning experiments and other theoretical work.

1. Referee question:

1. The Hall sign reversal phenomenon includes also a double sign change ... Can the proposed phenomenological consideration reveal the reason for that and explain this double change?

We answer:

Whether there are one or two sign reversals depends on the individual temperature dependence of each of the two terms in Eqn. (1). The first term (core conductivity) depends on the scattering time of the normal state quasiparticles in the vortex core. The second term, due to moving vortex charge (MVC) generally has the opposite sign, (see our answer to Q.4 below) and depends on the superfluid stiffness, which saturates at low temperatures. The (lower temperature) sign reversal reflects a rapid increase of core conductivity at low temperatures if it is dominated by inelastic (electron-electron and electron phonon) scattering. This point is clarified in Section 9.

2. Referee question:

According to the beautiful sketch of Figure 1 of the manuscript, the sign change is expected to occur below Bc2. However, there are many works (including, again, Ref. [7] of the manuscript,...reporting that the sign change starts already above Bc2. How should the sketch of Figure 1 reconcile with these observations?

We answer:

The curves in Fig. 1 depict Bardeen-Stephen mean field theory, which expects $B_{\rm c2}(T)$ to be sharp phase transition. Experimentally, $B_{\rm c2}$ is merely the midpoint in the magnetoresistivity rise between the flux flow and normal metal regimes. Evidently, short range phase correlations persist within a limited range above $T_c(B)$, as adduced from the broadened magnetoresistivity curves in Fig. (5) of the supplementary material of Zhao et.al. (Ref. [7]). A few layers of BSCCO films are expected to undergo a zero field Kosterlitz-Thouless (KT) transition. Halperin and Nelson (J. Low. Temp 36, 599 (1975)) theory describes mobile vortices in a limited range above $T_{\rm c}$. The MVC carried by these vortices can reverse the Hall sign even slightly above the nominal $T_{\rm c}(B)$. We discuss this point in Section 8.

3. Referee comment:

Reference [7] of the manuscript presents a quantitative description of the experimental data in the framework of vortex dynamics (Reference [15] of the manuscript). Based on the fair agreement between the theory and experiment, Reference [7] claims that the theory developed in [15] completely explains the phenomenon of the Hall sign change. One could then conclude that Ref. [15] of the manuscript together with the subsequent extension by Geshkenbein, Ioffe, and Larkin, Superconductivity in a system with preformed pairs, PRB 55, 3173 (1997), have already resolved all the controversies existing before. It would be useful for a reader to learn which deficiencies of these (Ref [15] and PRB of 1997) works were resolved in the present manuscript.

We answer:

The BSCCO data of Zhao et al., Ref. [7], was indeed fit by the theories of Feigel'man et al. in Ref. [15] and PRB 1997. However, our (different) theory also fits the data of Zhao et al., as well as that of Hagen et al. (Ref. [20]), which includes Hall anomalies in both hole and electron doped cuprates. That said, even good fits cannot prove any theory.

As mentioned in the introduction, and discussed now more thoroughly in Section 9, our paper addresses unresolved issues concerning the widely used vortex dynamics approach, which includes the theory of Ref. [15]. The old controversy between Bardeen and Stephen versus Nozieres and Vinen concerning the flux flow Hall effect has not yet been settled. We now understand that this controversy originates from an inherent shortcoming of the vortex dynamics equation. The vortex forces are chosen by phenomenological considerations, such as power dissipation and symmetries. A basic problem is that the equation is set up to calculate the generated EMF in response to a bias current. However, conductivities in the presence of potentials and interactions are feasibly calculated only by inserting an electric field into the Hamiltonian and evaluating the generated current. This is the approach we adopt in the present paper. The main advantage of this approach is that we can avoid the determination of vortex forces where momentum is not conserved. We do not need to calculate the vortex velocity from the vortex dynamics equation, since it is strictly constrained by Josephson equation to be ${\bf V} = c{\bf E} \times {\bf B}/B^2$, with no free parameters, as derived in Section 2. In addition, (as explained below) we derive the opposite sign of the vortex core charge than obtained by the Galilean symmetry consideration in Ref. [15].

4. Referee question:

... the authors apply the theory of the vortex screening by Khomskii and Freimuth (Ref. [10] of the manuscript). However, this theory gives the extra charge of the opposite sign than that obtained by other authors. How does this discrepancy may influence the conclusions of the manuscript?

We answer:

Feigel'man et al. (Ref. [15]) obtained a Hall conductivity contribution $\Delta\sigma_{xy} = -\delta n ec /B$. We refer to this specific result below as Eqn A. In Eqn. A, $\delta n = n_0 - n_{\infty}$ is the charge accumulated at the vortex core. Khomskii and Freimuth (KF), along with our work, arrived at the opposite conclusion, i.e. that Hall sign reversal is related to vortex core charge depletion rather than accumulation. Ref. [15], and references therein invoked Galilean symmetry which is supposed to be recovered for zero core charge. In that limit Eqn A yields $\sigma_{xy} = n_\infty ec /B$.

However, we dispute the validity of Eqn A. In Section 9 and Appendix B we explain why Galilean symmetry is not applicable to the flux flow regime of general superconductors, for two main reasons:

(1) A real superconductor is composed of band electrons in a periodic potential, and thus is not Galilean invariant. For example, Galilean symmetry dramatically fails near a half-filled band (e.g. at $n \approx 1$ for the square lattice tight binding model), where the Hall conductivity must vanish by particle-hole symmetry. de Gennes and Nozieres, in Phys. Lett. 15, 216 (1965), noted that Galilean fluid effects (e.g. vortex mutual precession) may be observable (perhaps) only for $\omega_c\tau \gg 1$. In BSCCO, $\omega_c \tau \sim 10^{-3}$ as shown by the Hall angle in our Fig. (6). These lie outside the Galilean effects regime.

(2) Even for a quadratic band, vortices will not "go with the flow", as expected by Kelvin's circulation theorem for classical fluids. The emergence of short range superconducting stiffness below $T_{\rm c}$, differentiates between the flux flow and normal metal transport. Even for parabolic bands, Galilean symmetry is broken in a superconductor due to rigidity of the vortex lattice pinned by arbitrary weak potentials. In the flux flow regime, where short range stiffness survives, we show in Appendix B that the vortices do not drag the background condensate. Hence the Hall conductivity is not given by $\sigma_{xy} = n_\infty ec /B$. Interestingly, it is given instead by the charge-flux attachment of the vortex liquid, i.e. the Streda formula which is derived from the Kubo formula in Appendix A: $\Delta\sigma_{xy} = ec {\partial n\over \partial B}$. This expression generally yields a sign opposite to that of Eqn. A.

A compelling phenomenological argument against Eqn. A, is that for BCS theory (as shown by KF), the vortex charge is lower than the uniform charge, i.e. $\delta n <0$. Hence Hall sign reversals which were observed in conventional superconductors, e.g. Nb and V by Noto et al. in Ref. [2], are not consistent with the positive sign of Eqn. A. As far as BSCCO is concerned, we hope that the sign of the vortex core charge may be directly measured by upcoming experiments in the near future.

---

## Round 2 · Referee Report · Anonymous · 2020-2-24

Strengths

1) The topic is old, but of considerable current interest, and even somewhat controversial.

2) The paper is well written and coherently argued.

Weaknesses

1) I do not find the key arguments to be totally convincing.

Report

The ideas in the paper build on the original Bardeen Stephen (B-S) model whose theoretical basis is phenomenological, and not fully convincing. In particular, when the superfuid is uncharged and the model Galilean invariant, a correct model should reproduce Kelvin's theorem that the vortices move with the flow. This property is preserved by Noziers and Vinen and by the time dependent Landau Ginzburg model considered by Dorsey, but not by B-S. Also the origin of the key equation ${\mathcal E}= -(\hbar/2e)\nabla \dot \phi$is unclear. Usually one thinks of the superfluid fluid being accelerated by the EMF rather than it creating and EMF. Indeed as it stands the equation cannot be quite correct as the EMF should be a gauge invariant quantity and $\nabla \dot \phi$ is missing the ${\bf A}$ field required for gauge covariance. I always though that as the ${\bf B}$ field is being advected with the vortex we could take it to be ${\bf A}({\bf r}-{\bf v}_Lt)$ and it is the ${\bf E}= - \dot {\bf A}$ that ensures that the vortex carries its circulation with it.

Requested changes

1) I'm confident that the authors know what they mean by their equations, but it would help a naive reader such as myself if they expanded their introduction to gave a clearer explanation of what is going on --- with particular attention to gauge invariance and to questions such as what would the vortex velocity be in the neutral Galailean limit.

  • validity: ok
  • significance: -
  • originality: high
  • clarity: good
  • formatting: excellent
  • grammar: excellent

Author:  Assa Auerbach  on 2020-03-06  [id 753]

(in reply to Report 2 on 2020-02-24)

Response to Referee Report 2

We thank the referee for a thorough report, and for the clarifying questions.

1. Referee comment:

The ideas in the paper build on the original Bardeen Stephen (B-S) model whose theoretical basis is phenomenological, and not fully convincing. In particular, when the superfluid is uncharged and the model Galilean invariant, a correct model should reproduce Kelvin's theorem that the vortices move with the flow. This property is preserved by Nozieres and Vinen and by the time dependent Landau Ginzburg model considered by Dorsey, but not by B-S.

Our answer:

We have not followed Bardeen and Stephen's (BS) approach, although we have assumed their (microscopically justifiable) model of a metallic vortex core of radius $k_F\xi \gg 1$. Their vortex dynamics equation (as does Nozieres and Vinen's (NV)) suffers from several lingering problems on which we elaborate in our answer to Question 3 of Referee 1, and in our expanded discussion in the new Section 9.

Indeed, it is quite surprising that with our orthogonal (current response) approach to flux flow transport theory, we have recovered BS conductivities $\sigma_{xx}(B)$ and $\sigma_{xy}(B)$ as the first term in Eq. (1). Our paper contradicts NV's result for $\sigma_{xy}$ and for the Hall angle, which we believe is due to invalid assumptions in their vortex forces. In our response to Question 4 of Referee 1, we explain why the Galilean invariance argument put forth by NV and adopted by many subsequent authors, is not relevant to flux flow regime of real superconductors. The aforementioned "go with the flow'' rule (Kelvin's circulation theorem for classical fluids), and therefore also the equation $\sigma_{xy} = nec/B$, are invalidated by the periodic lattice potential, impurities, and most importantly by the emergence of short range superfluid stiffness below $B_{\rm c2}$. Superconducting stiffness distinguishes the flux flow from the normal metal regimes, and has a different effect on the Hall conductivity than derived from Galilean symmetry arguments. This point is now well explained in Appendix B.

2. Referee comment:

The origin of the key equation $\varepsilon= - \hbar/(2e) \nabla \dot{\phi}$ is unclear. Usually one thinks of the superfluid fluid being accelerated by the EMF rather than it creating and EMF. Indeed as it stands the equation cannot be quite correct as the EMF should be a gauge invariant quantity and $\nabla\dot{\phi}$ is missing the ${\bf A}$ field required for gauge covariance.

Our answer:

The current density is gauge invariant ${\bf j} = \rho_s (\nabla \phi - {2e\over \hbar c} {\bf A})$. Throughout the paper we choose the gauge $A_0({\bf x},t) =0$ to define the external electric field as ${\bf E} = - {1\over c} \dot{\bf A}$ and vorticity produced EMF $\varepsilon= - \hbar/(2e) \nabla \dot{\phi}$. We apologise for previously neglecting to state our gauge choice, which is now stated after Eq. [4], and thank the Referee for the comment.

3. Referee comment:

I always thought that as the $B$ field is being advected with the vortex we could take it to be ${\bf A}({\bf r}- {\bf v}_L t)$, and it is the ${\bf A}= - \dot{\bf E}$ which ensures that the ${\bf A}$ field carries the circulation with it.

Our answer:

We beg to differ. The external magnetic field ${\bf B}$ is not advected, because it is approximately uniform in the (unscreened) thin film where $\lambda \gg l_B$. Therefore, the vector potential ${\bf A}_B = -{1\over 2} {\bf r}\times {\bf B}$ is time independent. The motion of vortices in the steady state is dictated by a uniform external electric field ${\bf E}$ which we introduce with ${\bf A}_E = -c {\bf E}t$. The vortices in the uniform (DC current) steady state must move at the Josephson velocity ${\bf V} = c {\bf E} \times {\bf B}/B^2$, as derived in Section 2.

---

## Round 3 · Referee Report · Anonymous · 2020-3-7

Report
The authors adequately replied to my comments. And although some further counterarguments can be raised, I find that this further scientific discussion should be exercised in the format of the whole interested community and that the reviewer must not delay the publication on the ground of arguable comments. As I stated before, the paper is well written and will definitely stimulate further progress in the field. I thus recommend publication of the manuscript.
Anonymous on 2020-03-09 [id 755]
What is the justification for treating the vortex core as a uniform & continuous metal when the coherence length is of the order of the lattice spacing?
Anonymous on 2020-03-09 [id 756]
(in reply to Anonymous Comment on 2020-03-09 [id 755])You are certainly right that the large metallic core model is unjustified for cuprates at low temperatures, when $k_F\xi \sim O(1)$. That is why we restricted our experimental analysis of BSCCO in Section 7 to $T> 65 K$, where the coherence length satisfies $k_F \xi(T) >k_F l > 1$. Therefore near $T_c$, our extrapolation of the Hall angle (Fig. 6) from above $T_c$ is justified. The moving vortex charge Hall conductivity is only weakly dependent on the vortex core via the parameter $B_{c_2}(T)$.

---

## Round 3 · Referee Report · Anonymous · 2020-3-9

Strengths
A potentially mportant contribution to a confused field.
Weaknesses
It is not at all clear in this paper what is being $assumed$ in the physics, and what is somewhere (but not in the text) $derived$ from assumptions. This
makes the paper hard to understand.
Report
I do not want to delay publication, but the paper really would be be improved
if some further explanation of the core ideas were given.
The key equation $ {\mathcal E}=- {\hbar/2e} \nabla\dot \phi$ is neither derived, explained, or cited from
elsewhere. Is it a definition of ${\mathcal E}$? If so, how does ${\mathcal E}$ act as a true EMF? If it is a genuine electric field which Maxwell equation does it come from? Or is it
an electrochemical potential of some sort? Does ${\mathcal E}$ drive the time evolution of the phase or does the time evolution of the phase somehow create ${\mathcal E}$?
I can see ${\mathcal E}$ coming from ${\mathcal E}= -\dot {\bf A}/c$ and
$$
{\bf j}\propto \left(\nabla \phi - \frac {2e}{\hbar c} {\bf A}\right)
$$
if we assert that
$$
{\bf j}_v\equiv {\rm const.} \left(\nabla \phi_v - \frac {2e}{\hbar c} {\bf A}\right)
$$
is pointwise constant, but why should this be the case?
Further this relation cannot universally true because dissipative processes can allow vortices to move against the flow even in neutral superfluids. In that
case the time evolution of $\phi$ is driven by the contribution to the Josephson
phase evolution equation from the interplay between appplied current and the vortex current in the gradient-squared kinematic part of the
energy. Are we ignoring such effects?
All this is particularly interesting because of equation 9. This is a mathematical identity given the definition of ${\mathcal E}$ and $\phi_v$. It also looks like Josephson's
relation between the electric field and the vortex velocity, but Josephson explicitly assumes that the moving flux lines lead to a time dependent magnetic
field and so his EMF is an genuine electric field from Faraday's flux rule. In their response to my initial review the authors say that the ${\bf B}$ field is not being advected with the vortex flow. So what generates the EMF?
Eq 9 has a trivial typo: The first equality has $\bar {\mathcal E}$ parallel to ${\bf V}$ while the last
one has it perpendicular. A $\hat z \times$ is missing, I think.
Author: Assa Auerbach on 2020-03-12 [id 763]
(in reply to Report 2 on 2020-03-09)
Response to Second Referee Report 2
We thank the referee for the followup report. We clarify the remaining issues below.
1. Referee comment:
It is not at all clear in this paper what is being assumed in the physics and what is somewhere (but not in the text) derived from assumptions. This makes the paper hard to understand.
Our answer:
Our assumptions (stated in the text) are: (i) The existence of a vortex liquid regime between the vortex melting field and normal metal regime, as written in Eq. (3). (ii) The Ginzburg-Landau free energy (Eq. (21)) governs this regime. (iii) The set up for transport calculations is in Fig. 3. The electric field is externally determined, and the current can flow freely in both longitudinal and transverse directions. This is realized in a Corbino geometry - the geometry of conductivity measurements. (iv) For the moving vortex charge effect (and Hall anomaly) we require a layered structure of low mobility dopant layers between superconducting planes, shown in Fig. 5.
2. Referee question:
Is $\varepsilon = -\hbar/2e \nabla\dot{\phi}$ a definition of $\varepsilon$? Is it a genuine electric field ... or is it an electrochemical potential of some sort? Does the time evolution of the phase somehow create $\varepsilon$? why should that be the case that ${\bf j}_v={\rm const}$?
Our answer:
Yes, $\varepsilon = -\hbar/2e \nabla\dot{\phi}$ is a definition. It is not the electric field, but the hydrodynamical (chemical potential ) stress field due to moving vortices. It is ab manifestation of Jospehson's relation. Only in the flux flow steady state $ \langle \varepsilon\rangle = {\bf E}$, where ${\bf E}$ is the external electric field. This equality allows ${\bf j} ={\rm const}$. In the pinned vortices phase, for example, $0=\varepsilon\ne {\bf E}$, and an instability occurs because the energy and current increase with time.
3. Referee question:
The time evolution of $\phi$ is driven by... the interplay between applied current and the vortex current .... Are we ignoring such effects?
Our Answer:
In our flux flow transport theory we do not consider a bias current, but an imposed electric field. The values of $\dot{\phi},{\bf V}, \varepsilon$ are constrained (in the steady state) by ${\bf E}$. This eliminates uncertainties about vortex forces and interplay between them. We are not neglecting any effects, its just that our approach does not require us to define them.
4. Referee question:
Equation 9 ... looks like Josephson's relation between the electric field and the vortex velocity, but Josephson explicitly assumes that the moving flux lines lead to a time dependent magnetic field and so his EMF is a genuine electric field from Faraday's flux rule. In their response to my initial review the authors say that ${\bf B}$ field is not being advected with the vortex flow. So what generates the EMF?
Our answer:
Indeed Josephson Phys. Lett. 1965 paper explains ${\bf E} =-{1\over c} {\bf V} \times {\bf B}$ as if produced by motion of flux tubes, as the Referee correctly points out. However, its been long realized that this equation holds even when $\lambda \gg l_B$, and the magnetic field is nearly uniform in space. This is especially true in thin films, where $\lambda$ can exceed the dimensions of the film. The general justification of Josephson's equation is due to the hydrodynamical stress field induced by moving vorticity. Vorticity and flux tubes only move together in the strong screening $\lambda << l_B$ regime. Motion of vorticity however always induces a transverse pressure gradient, even in neutral superfluids. In normal fluids it is related to "Crocco's theorem".

---

## Round 3 · Author Response

We hereby resubmit our revised Manuscript entitled "Hall anomaly and moving vortex charge in layered superconductors", by Assa Auerbach and Daniel P. Arovas. In response to reports of Referees 1 and 2, this version incorporates the answers to their questions. The discussion section has been significantly expanded to explain the advantages of our approach, where the current is derived by imposing an external electric field into the Ginzburg-Landau Hamiltonian. We show how this approach resolves longstanding controversies related to determination of vortex forces and dynamics. A Galilean symmetry argument, which was invoked by previous papers to determine the flux-flow Hall conductivity, is refuted in Appendix B.

---

## Round 3 · List of Changes

1. We add a comment in the abstract about the content of the two appendices.
2. Section 2: After Eq. (4) we add a note about fixing the gauge by setting A_0=0.
3. Section 8: Last paragraph adds a discussion of the reason for experimentally observed Hall sign reversal due to vortices existing even slightly above the nominal T_c.
4. Section 9: The expanded discussion section now includes: a) The inherent difficulty of setting up a microscopic basis for vortex dynamics equations. b) The invalidity of Galilean symmetry arguments for the Hall effect, as applied to the flux flow regime of real superconductors. c) An explanation of two Hall sign reversals as effects of inelastic scattering effecting the vortex core conductivity.
5. Appendix B: A new appendix which addresses a log time misconception in the superconducting literature. We prove by three complementary arguments, that even for Galilean invariant Hamiltonians, short range superconducting stiffness prevents the vortices from "going with the flow" as expected by Kelvin's theorem for classical liquids.

---

## Editorial Decision

published